# A New Nonlinear Spatial Compliance Model Method for Flexure Leaf Springs with Large Width-to-Length Ratio under Large Deformation

**DOI:** 10.3390/mi13071090

**Published:** 2022-07-09

**Authors:** Yin Zhang, Jianwei Wu, Jiansheng Pan, Zhenzhuo Yan, Jiubin Tan

**Affiliations:** 1Centre of Ultra-Precision Optoelectronic Instrumentation Engineering, Harbin Institute of Technology, Harbin 150001, China; 17b901008@stu.hit.edu.cn (Y.Z.); 22b301002@stu.hit.edu.cn (J.P.); 21b901042@stu.hit.edu.cn (Z.Y.); jbtan@hit.edu.cn (J.T.); 2Key Lab of Ultra-Precision Intelligent Instrumentation, Harbin Institute of Technology, Ministry of Industry and Information Technology, Harbin 150080, China

**Keywords:** flexure leaf spring, large width-to-length ratio, nonlinear compliance model, chain model, parametric analysis

## Abstract

Flexure leaf spring (FLS) with large deformation is the basic unit of compliant mechanisms with large stroke. The stiffness along the non-working directions of FLSs with large width-to-length ratio (*w*/*L*) is high. The motion stability of the compliant mechanism based on this type of FLS is high. When this type of FLS is loaded along the width direction, the shear deformation needs to be characterized. Nevertheless, currently available compliance modeling methods for FLS are established based on Euler–Bernoulli beam model and cannot be used to characterize shear models. Therefore, these methods are not applicable in this case. In this paper, a new six-DOF compliance model for FLSs with large *w*/*L* is established under large deformation. The shear deformation along the width direction model is characterized based on the Timoshenko beam theory. The new constraint model and differential equations are established to obtain a high-precision compliance model expression for this type of FLS. The effects of structural parameters on the compliance of the FLS are analyzed. Finally, the accuracy of the model is verified both experimentally and by finite element simulation. The relative error between theoretical result and experiment result is less than 5%.

## 1. Introduction

Compliant mechanisms are mechanical mechanisms that rely on the elastic deformation of materials and, among other uses, have been widely applied in medical devices [1,2,3], precision instruments [4,5,6], sensors [7,8,9], and microelectronic-mechanical systems (MEMS) [10,11]. Compliant mechanisms can be divided into centralized and distributed compliant mechanisms. The deformation of centralized compliant mechanisms is only concentrated at flexure hinges under small deformation. The compliance of the flexure hinges and the overall centralized compliant mechanism can be modelled based on the premise of linear deformation [12,13,14]. On the other hand, distributed compliant mechanisms are based on the deformation of the entire structure or most of it. In the same design space, the stroke of the distributed compliant mechanism is larger than that of the centralized one; thus, distributed compliant mechanisms have been widely used in precision machinery and precision instruments [15,16,17]. An accurate compliance model is the basis for the structural design of such mechanisms. When large deformation occurs in a distributed compliant mechanism, there is geometric nonlinearity in the flexure leaf spring (FLS). To this end, it is important to investigate and develop a nonlinear compliance model with high accuracy able to simulate the large deformation of FLS.

Initially, nonlinear compliance models concerned FLSs with 2D planar deformation. The methods used include the beam constraint method [18], chain model method [19], elliptic integral method [20], and pseudo-rigid-body method [21,22]. The beam constraint method is derived from the basic equation in the elastic mechanics theory and its accuracy is high when the deformation range of the FLS is small, i.e., less than 0.1 times the length. For example, Radgolchin et al. [23] proposed a nonlinear static model with an intermediate semi-rigid element of load–displacement relationships of flexure modules based on FLSs. However, the beam constraint method cannot be applied in cases of large deformation. In the chain model method, accurate models of units with small and precise deformations are assembled and analyzed. Chen et al. [24] proposed a version of the chain method to model large planar deflections of initially curved beams. This method can be easily adapted to FLSs of various shapes. The elliptic integral method can provide high precision for large deformations. Wang et al. [25] proposed a new design of a flexure-based XY precision positioning stage and analyzed its characteristics using the elliptic integral method. Finally, the pseudo-rigid-body method is an efficient and easy-to-understand analysis method. In this method, the deformation and stiffness of the FLS with large deformation is simulated through connected rigid rods and torsion springs at the connections. For instance, based on the pseudo-rigid-body model, Yu et al. [26] proposed a new model with three degrees-of-freedom (DOFs) for FLSs with large deflection.

The 2D nonlinear compliance models of FLS are only suitable for planar distributed compliant mechanisms. Therefore, 3D compliance models are needed for the modelling of spatial multi-DOF distributed compliant mechanism. Irschik et al. [27] proposed a continuum mechanics-based interpretation of Reissner’s structural mechanics model, where a proper continuum mechanics-based meaning is attached to both the generalized static entities and strains in Reissner’s theory. Sen et al. [28] analyzed the constraint characteristics of a uniform and symmetric cross-sectioned, slender, spatial beam, which can used in 3D compliant mechanisms. Brouwer et al. [29] presented refined analytic equations for the stiffness in three dimensions taking into account the shear compliance, constrained warping, and limited parallel external drive stiffness. Nijenhuis et al. [30] presented a modeling approach for obtaining insight into the deformation and stiffness characteristics of general 3D flexure strips that undergo bending, shear, and torsion deformation. Bai et al. [31,32,33] proposed load–displacement relationships for rectangular and large-aspect-ratio beams by solving their nonlinear governing differential equations using the power series method.

Highly deformable FLSs with low stiffness along the working direction and high stiffness in the non-working directions have been widely used, e.g., in large-stroke compliant linear guiding mechanisms. In general, the width-to-length ratio (*w*/*L*) of this type of FLSs is large. Current 3D compliance models all can be used in FLSs with small width-to-length ratio and thickness-to-length ratio. When establishing a compliance model for this type of FLS, the constraint model based on Euler–Bernoulli beam theory and the spatial coordinate system conversion relationship are used to construct a differential equation system without considering shear deformation. Therefore, current 3D compliance models cannot be used to accurately predict the compliance of such FLSs. To this end, this paper proposes a spatial compliance model for FLSs with large *w*/*L*. In this model, the shear deformation along the width direction is characterized based on the Timoshenko beam theory and a new differential equation system is established by combining the spatial coordinate transformation relations. The shear deformation along the width direction of the FLS with large *w*/*L* and the spatial geometric nonlinearity under large deformation can be accurately described by this model. Furthermore, an experimental platform is set to verify the accuracy of the proposed compliance model.

The rest of the paper is organized as follows. In Section 2, a compliance model of the FLS with a small deformation (<0.1 *L*) is established based on the spatial constraint model and the relationship between deformation and load is derived. In Section 3, a spatial chain model is established. In particular, a spatial six-DOF compliance model of the FLS is obtained by connecting FLSs with small deformation based on the coordinate transformation method. In Section 4, the effects of the structural parameters on FLS compliance and compliance ratio are analyzed. In Section 5, an experimental platform is built to verify the accuracy of the model proposed in this paper.

## 2. Six-DOF Compliance Model of the FLS Element under Small Deformation

### 2.1. Expressions of Deformation and Load

The deformation of the FLS under load and the definition of the structural parameters are given in Figure 1. The position of the point P1 on the FLS is expressed as (*X*, 0, 0) when the FLS is not deformed and as *P*_1_′ (*X* + *U*_X_, *U*_Y_, *U*_Z_) after deformation. It is defined by the cross-section where the point *P*_1_ (point *P*_1_′ after deformation) is located perpendicular to the tangential direction of the FLS axis at *P*_1_′ after deformation. A coordinate system is established on the cross-section, where the point *P*_1_′ is located after deformation. The *X*_d1_ axis of the coordinate system coincides with the tangent of the axis at *P*_1_′, and when the FLS is not deformed, the directions of the *Y*_d1_ and *Z*_d1_ axes are the same as those of the Y and Z axes of the initial coordinate system. The transformation relationship between the initial coordinate system (*X*, *Y*, *Z*) and the one on the cross-section where the point P_1_′ is located after deformation (*X*_d1_, *Y*_d1_, *Z*_d1_) is depicted in Figure 2 and it can be expressed as:(1)$$\begin{array}{l} R_{d}=R_{x}\left(\Theta_{\mathrm{Xd}}\right)R_{y}\left(\Theta_{Y}\right)R_{z}\left(\Theta_{Z}\right)\\ =\left[\begin{array}{ccc} 1 & 0 & 0\\ 0 & c\left(\Theta_{\mathrm{Xd}}\right) & -s\left(\Theta_{\mathrm{Xd}}\right)\\ 0 & s\left(\Theta_{\mathrm{Xd}}\right) & c\left(\Theta_{\mathrm{Xd}}\right) \end{array}\right]\left[\begin{array}{ccc} c\left(\Theta_{Y}\right) & 0 & s\left(\Theta_{Y}\right)\\ 0 & 1 & 0\\ -s\left(\Theta_{Y}\right) & 0 & c\left(\Theta_{Y}\right) \end{array}\right]\left[\begin{array}{ccc} c\left(\Theta_{Z}\right) & -s\left(\Theta_{Z}\right) & 0\\ s\left(\Theta_{Z}\right) & c\left(\Theta_{Z}\right) & 0\\ 0 & 0 & 1 \end{array}\right]\\ =\left[\begin{array}{ccc} c\left(\Theta_{Y}\right)c\left(\Theta_{Z}\right) & c\left(\Theta_{Z}\right)s\left(\Theta_{\mathrm{Xd}}\right)c\left(\Theta_{Y}\right)-c\left(\Theta_{\mathrm{Xd}}\right)s\left(\Theta_{Z}\right) & s\left(\Theta_{Z}\right)s\left(\Theta_{\mathrm{Xd}}\right)+c\left(\Theta_{\mathrm{Xd}}\right)c\left(\Theta_{Z}\right)s\left(\Theta_{Y}\right)\\ c\left(\Theta_{Y}\right)s\left(\Theta_{Z}\right) & c\left(\Theta_{Z}\right)c\left(\Theta_{\mathrm{Xd}}\right)+s\left(\Theta_{\mathrm{Xd}}\right)s\left(\Theta_{Z}\right)s\left(\Theta_{Y}\right) & c\left(\Theta_{\mathrm{Xd}}\right)s\left(\Theta_{Y}\right)s\left(\Theta_{Z}\right)-c\left(\Theta_{Z}\right)s\left(\Theta_{\mathrm{Xd}}\right)\\ -s\left(\Theta_{Y}\right) & c\left(\Theta_{Y}\right)s\left(\Theta_{\mathrm{Xd}}\right) & c\left(\Theta_{\mathrm{Xd}}\right)c\left(\Theta_{Y}\right) \end{array}\right] \end{array}$$
where c(Θ)=cos(Θ), s(Θ)=sin(Θ).

When the deformation of the FLS is small, i.e., less than 0.1 *L*, the following approximate relationship can be satisfied [31]:(2)$$\{\begin{array}{l} \Theta_{Y}\approx -U_{Z}^{\prime }\;\\ \Theta_{Z}\approx U_{Y}^{\prime } \end{array}$$

The torsion around the *X* axis and the curvature around the *Y* and *Z* axes can be expressed as [28]:(3)$$\{\begin{array}{l} \kappa_{X}=\Theta_{\mathrm{Xd}}^{\prime }+\Theta_{Y}^{\prime }\Theta_{Z}^{\prime }=\Theta_{\mathrm{Xd}}^{\prime }-U_{Z}^{\prime \prime }U_{Y}^{\prime }\\ \kappa_{Y}=\Theta_{Y}^{\prime }+\Theta_{\mathrm{Xd}}^{\prime }\Theta_{Z}^{\prime }=-U_{Z}^{\prime \prime }+\Theta_{\mathrm{Xd}}^{\prime }U_{Y}^{\prime \prime }\\ \kappa_{Z}=\Theta_{Z}^{\prime }-\Theta_{\mathrm{Xd}}^{\prime }\Theta_{Y}^{\prime }=U_{Y}^{\prime \prime }+\Theta_{\mathrm{Xd}}U_{Z}^{\prime \prime } \end{array}$$

The loads on the end of the FLS are *F*_XL_, *F*_YL_, *F*_ZL_, *M*_XL_, *M*_YL_, and *M*_ZL_. The deformation and rotation angles at the point PL at the end of the FLS are expressed as *U*_X1_, *U*_Y1_, *U*_Z1_, and Θ_Xd1_, Θ_Y1_, Θ_Z1_, respectively. The load applied to any point on the FLS can be expressed as (4).
(4)$$\{\begin{array}{l} F_{X}=F_{\mathrm{XL}}\\ F_{Y}=F_{\mathrm{YL}}\\ F_{Z}=F_{\mathrm{ZL}}\\ M_{X}=M_{\mathrm{XL}}-F_{\mathrm{ZL}}(U_{\mathrm{YL}}-U_{Y})-F_{\mathrm{YL}}(U_{\mathrm{ZL}}-U_{Z})\\ M_{Y}=M_{\mathrm{YL}}-F_{\mathrm{XL}}(U_{\mathrm{ZL}}-U_{Z})-F_{\mathrm{ZL}}(L+U_{X}-X)\\ M_{Z}=M_{\mathrm{ZL}}+F_{\mathrm{YL}}(L+U_{X}-X)-F_{\mathrm{XL}}(U_{\mathrm{YL}}-U_{Y}) \end{array}$$

### 2.2. Differential Governing Equations

This paper considers that the *w*/*L* of the FLS is large. This section analyzes the calculation process of the deformation of every DOF.

#### 2.2.1. Deformation in the *Z*-*O*-*X* Plane

The loads in the *Z*-*O*-*X* plane are *F*_ZL_ and *M*_YL_. Due to the *F*_ZL_, there is shear deformation in the FLS width direction, while, due to the *M*_YL_, there is deflection deformation. The deformation in the *Z*-*O*-*X* plane can be expressed based on the Timoshenko beam theory as follows:(5)$$\{\begin{array}{l} M_{Y}=EI_{Y}\kappa_{Y}\\ \frac{dU_{Y}}{dX}=\frac{1}{k^{2}AG}\frac{dM_{Y}}{dx}-\int \kappa_{Y}dX \end{array}$$
where *k*^2^ is the shear coefficient, here *k*^2^ = 0.85; *E* is the elastic modulus; *G* is the shear modulus; *A* is cross-sectional area of the FLS, which is defined as *A* = *wt*; and *I*_Y_ is the inertia moment around the *Y* axis, which is defined as *I*_Y_ = *w*^3^*t*/12.

#### 2.2.2. Deformation in the *Y*-*O*-*X* Plane

The loads in the *Y*-*O*-*X* plane are *F*_YL_ and *M*_ZL_, due to which there is deflection deformation in the FLS thickness direction. The deformation in the *Y*-*O*-*X* plane can be expressed based through the Euler–Bernoulli beam theory as follows:(6)$$M_{Z}=EI_{Z}\kappa_{Z}$$
where *I*_Z_ is the inertia moment around the *Z* axis, which is defined as *I*_Z_ = *wt*^3^/12.

#### 2.2.3. Torsion and Extension/Contraction

When the FLS is loaded by *M*_X_, there is warping deformation due to the effect of shear force [30]. The deformation can be expressed by the following equation:(7)$$M_{X}=GJ\kappa_{X}$$
where *J* is the torsional constant of the FLS and can be expressed as [34]:(8)$$J=\frac{2t^{3}w^{3}}{7t^{2}+7w^{2}}f\left(\eta \right)$$
where *η* = *t*/*w*. The expression of f(*η*) is as follows:(9)$$f\left(\eta \right)=\frac{1.167\eta^{5}+29.49\eta^{4}+30.9\eta^{3}+100.9\eta^{2}+30.38\eta +29.41}{\eta^{5}+25.91\eta^{4}+41.58\eta^{3}+90.43\eta^{2}+41.74\eta +25.21}$$

The relationship between the expansion/contraction deformation of the FLS and *F*_X_ can be expressed as follows [31]:(10)$$\begin{array}{ll} F_{\mathrm{XL}} & =E\iint_{A}\varepsilon_{\mathrm{XX}}dA\\ & =E\int_{-t/2}^{t/2}\int_{-w/2}^{w/2}\left(U_{X}^{\prime }+\frac{1}{2}U_{Y}^{\prime 2}+\frac{1}{2}U_{Z}^{\prime 2}-t\kappa_{Z}+w\kappa_{Y}+\frac{1}{2}\kappa_{X}^{2}\left(t^{2}+w^{2}\right)\right)dYdZ\\ & =EA\left(U_{X}^{\prime }+\frac{1}{2}U_{Y}^{\prime 2}+\frac{1}{2}U_{Z}^{\prime 2}\right)+\frac{E\left(I_{Y}+I_{Z}\right)}{2}\kappa_{X}^{2}\\ & \approx EA\left(U_{X}^{\prime }+\frac{1}{2}U_{Y}^{\prime 2}+\frac{1}{2}U_{Z}^{\prime 2}\right) \end{array}$$

It should be noted that $\frac{E\left(I_{Y}+I_{Z}\right)}{2}\kappa_{X}^{2}$ is so small that it can be ignored.

#### 2.2.4. Global System of Equations

The global system of differential governing equations for the spatial FLS is as follows:(11)$$\{\begin{array}{l} M_{\mathrm{YL}}-F_{\mathrm{XL}}(U_{\mathrm{ZL}}-U_{Z})=EI_{Y}\left(-U_{Z}^{\prime \prime }+\Theta_{\mathrm{Xd}}^{\prime }U_{Y}^{\prime \prime }\right)\\ U_{Y}^{\prime }=\frac{F_{\mathrm{ZL}}}{k^{2}AG}-\int \left(-U_{Z}^{\prime \prime }+\Theta_{\mathrm{Xd}}^{\prime }U_{Y}^{\prime \prime }\right)dX\\ M_{\mathrm{ZL}}+F_{\mathrm{YL}}(L-X)-F_{\mathrm{XL}}(U_{\mathrm{YL}}-U_{Y})=EI_{Z}\left(U_{Y}^{\prime \prime }+\Theta_{\mathrm{Xd}}U_{Z}^{\prime \prime }\right)\\ M_{\mathrm{XL}}-F_{\mathrm{ZL}}(U_{\mathrm{YL}}-U_{Y})-F_{\mathrm{YL}}(U_{\mathrm{ZL}}-U_{Z})=GJ\left(\Theta_{\mathrm{Xd}}^{\prime }-U_{Z}^{\prime \prime }U_{Y}^{\prime }\right)\\ F_{\mathrm{XL}}=EA\left(U_{X}^{\prime }+\frac{1}{2}U_{Y}^{\prime }+\frac{1}{2}U_{Z}^{\prime }\right) \end{array}$$

The load and deformation can be normalized as follows:(12)$$\left\{ \begin{array}{l} f_{X1}=\frac{F_{\mathrm{XL}}L^{2}}{EI_{Z}},f_{Y1}=\frac{F_{\mathrm{YL}}L^{2}}{EI_{Z}},f_{Z1}=\frac{F_{\mathrm{ZL}}L^{2}}{EI_{Z}},m_{X1}=\frac{M_{\mathrm{XL}}L}{EI_{Z}}\\ m_{Y1}=\frac{M_{\mathrm{YL}}L}{EI_{Z}},m_{Z1}=\frac{M_{\mathrm{ZL}}L}{EI_{Z}},u_{Y1}=\frac{U_{\mathrm{YL}}}{L},u_{Z1}=\frac{U_{\mathrm{ZL}}}{L}\\ u_{X}=\frac{U_{X}}{L},u_{Y}=\frac{U_{Y}}{L},u_{Z}=\frac{U_{Z}}{L},\theta_{\mathrm{Xd}}=\Theta_{\mathrm{Xd}},x=\frac{X}{L} \end{array}\right.$$
where
(13)$$\{\begin{array}{l} \alpha =\frac{I_{Y}}{I_{Z}},\beta =\frac{1}{EI_{Z}},\gamma =\frac{1}{k^{2}AG}\\ \lambda =\frac{GJ}{EI_{Z}},\xi =\frac{AL^{2}}{I_{Z}} \end{array}$$

The system of equations can be expressed as:(14)$$\{\begin{array}{l} m_{Y1}-f_{X1}(u_{Z1}-u_{Z})=\alpha \left(-u_{Z}^{\prime \prime }+L\theta_{\mathrm{Xd}}^{\prime }U_{Y}^{\prime \prime }\right)\\ \beta u_{Y}^{\prime }=\gamma f_{Z1}-\beta \int \left(-u_{Z}^{\prime \prime }+L\theta_{\mathrm{Xd}}^{\prime }U_{Y}^{\prime \prime }\right)dX\\ m_{Z1}+f_{Y1}(1-x)-f_{X1}(u_{Y1}-u_{Y})=\left(u_{Y}^{\prime \prime }+L\theta_{\mathrm{Xd}}u_{Z}^{\prime \prime }\right)\\ m_{X1}-f_{z1}(u_{Y1}-u_{Y})-f_{y1}(u_{Z1}-u_{Z})=\lambda \left(\theta_{\mathrm{Xd}}^{\prime }-Lu_{Z}^{\prime \prime }u_{Y}^{\prime \prime }\right)\\ f_{X1}=\xi \left(u_{X}^{\prime }+\frac{1}{2}Lu_{Y}^{\prime \prime 2}+\frac{1}{2}Lu_{Z}^{\prime \prime 2}\right) \end{array}$$

### 2.3. Differential Governing Equations

In Figure 1, the FLS is fixed at *O*; thus, the boundary conditions for the displacement of the FLS can be expressed as:(15)$$\left\{ \begin{array}{l} U_{X}(0)=0,U_{Y}(0)=0,U_{Z}(0)=0\\ \Theta_{X}(0)=0,U_{Y}^{\prime }(0)=0,U_{Z}^{\prime }(0)=0 \end{array}\right.$$

Every deformation of the FLS is derivable from the axial coordinate *X* and can be expressed as a power series of *X*. Therefore, the solution method based on series is employed to solve the system of equations. Every deformation can be expressed as:(16)$$\{\begin{array}{l} u_{X}=\sum_{n=0}^{\infty }a_{n}x^{n}\\ \begin{array}{l} u_{Y}=\sum_{n=0}^{\infty }b_{n}x^{n}\\ u_{Z}=\sum_{n=0}^{\infty }c_{n}x^{n}\\ \theta_{\mathrm{Xd}}=\sum_{n=0}^{\infty }d_{n}x^{n} \end{array} \end{array}$$
where *x* = *X*/*L*. The expressions of *θ*_Y_ and *θ*_Z_ can be obtained according to Equation (2) as follows:(17)$$\{\begin{array}{l} \theta_{Y}=-u_{Z}^{\prime }=-\sum_{n=1}^{\infty }\left(n-1\right)c_{n}x^{n-1}\\ \theta_{Z}=u_{Y}^{\prime }=-\sum_{n=1}^{\infty }\left(n-1\right)b_{n}x^{n-1} \end{array}$$

The following system of equations can be obtained by substituting Equation (16) into Equation (11):(18)$$\{\begin{array}{l} m_{Y1}-f_{X1}(\sum_{n=0}^{\infty }c_{n}-\sum_{n=0}^{\infty }c_{n}x^{n})=\alpha \left(-\sum_{n=2}^{\infty }n\left(n-1\right)c_{n}x^{n-2}+L\left(\sum_{n=1}^{\infty }nd_{n}x^{n-1}\right)\left(\sum_{n=2}^{\infty }n\left(n-1\right)b_{n}x^{n-2}\right)\right)\\ \beta \sum_{n=1}^{\infty }nb_{n}x^{n-1}=\gamma f_{Z1}-\beta \int \left(-\sum_{n=2}^{\infty }n\left(n-1\right)c_{n}x^{n-2}+L\left(\sum_{n=1}^{\infty }nd_{n}x^{n-1}\right)\left(\sum_{n=2}^{\infty }n\left(n-1\right)b_{n}x^{n-2}\right)\right)dX\\ m_{Z1}+f_{Y1}(1-x)-f_{X1}(\sum_{n=0}^{\infty }b_{n}-\sum_{n=0}^{\infty }b_{n}x^{n})=L^{2}\left(\sum_{n=2}^{\infty }n\left(n-1\right)b_{n}x^{n-2}+L\left(\sum_{n=0}^{\infty }d_{n}x^{n}\right)\left(\sum_{n=2}^{\infty }n\left(n-1\right)c_{n}x^{n-2}\right)\right)\\ m_{x1}-f_{z1}(\sum_{n=0}^{\infty }b_{n}-\sum_{n=0}^{\infty }b_{n}x^{n})-f_{y1}(\sum_{n=0}^{\infty }c_{n}-\sum_{n=0}^{\infty }c_{n}x^{n})=\lambda \left(\sum_{n=1}^{\infty }nd_{n}x^{n-1}-L\left(\sum_{n=2}^{\infty }n\left(n-1\right)c_{n}x^{n-2}\right)\left(\sum_{n=1}^{\infty }nb_{n}x^{n-1}\right)\right)\\ f_{X1}=\xi \left(\sum_{n=1}^{\infty }na_{n}x^{n-1}+\frac{1}{2}L{\left(\sum_{n=1}^{\infty }nb_{n}x^{n-1}\right)}^{2}+\frac{1}{2}L{\left(\sum_{n=1}^{\infty }nc_{n}x^{n-1}\right)}^{2}\right) \end{array}$$

The coefficients in Equation (16) are related to the load and structural parameters of the FLS. When Equation (18) is solved, the coefficients of the same order terms in the system of equations are made be equal to zero. In this paper, the numerical solution method is applied. The *a*_n_, *b*_n_, *c*_n_, and *d*_n_ coefficients gradually approach 0 with increasing n. Therefore, the result can be considered to be convergent. When the deformation of the FLS is small, the high-order terms in Equation (16) are smaller than the first- and second-order terms. Consequently, the fifth- and higher-order terms are ignored. The final result is obtained using the polynomial fitting method and is expressed as Equation (19).
(19)$$\left\{ \begin{array}{l} u_{Y1}=c_{Y1}f_{Y1}+c_{Y2}m_{Z1}+c_{Y3}f_{Z1}+c_{Y4}m_{Y1}\\ \theta_{Y1}=c_{Y11}f_{Y1}+c_{Y22}m_{Z1}+c_{Y33}f_{Z1}+c_{Y44}m_{Y1}\\ u_{Z1}=c_{Z1}f_{Y1}+c_{Z2}m_{Z1}+c_{Z3}f_{Z1}+c_{Z4}m_{Y1}\\ \theta_{Z1}=c_{Z11}f_{Y1}+c_{Z22}m_{Z1}+c_{Z33}f_{Z1}+c_{Z44}m_{Y1}\\ u_{X1}=\frac{f_{X1}}{\xi }-\frac{m_{X1}^{2}}{\lambda^{2}\xi }+\left[\begin{array}{llll} f_{Y1} & m_{Z1} & f_{Z1} & m_{Y1} \end{array}\right]\left(c_{X1}+f_{X1}c_{X2}+m_{X1}c_{X3}\right)\left[\begin{array}{l} f_{Y1}\\ m_{Z1}\\ f_{Z1}\\ m_{Y1} \end{array}\right]\\ \theta_{X1}=\frac{m_{X1}}{\lambda }-\frac{2m_{X1}f_{X1}}{\xi^{2}\lambda }+\left[\begin{array}{llll} f_{Y1} & m_{Z1} & f_{Z1} & m_{Y1} \end{array}\right]\left(c_{X11}+f_{X1}c_{X22}+m_{X1}c_{X33}\right)\left[\begin{array}{l} f_{Y1}\\ m_{Z1}\\ f_{Z1}\\ m_{Y1} \end{array}\right] \end{array}\right.$$

The variable *h*_s_ is set as $h_{s}=\frac{\gamma E}{\xi A}$ The symbols and matrices in Equation (16) can be expressed as follows:(20)$$\left\{ \begin{array}{l} c_{Y1}=-g_{22}g_{33}g_{44}+g_{22}g_{34}g_{43}+g_{23}g_{32}g_{44}-g_{23}g_{34}g_{42}-g_{24}g_{32}g_{43}+g_{24}g_{33}g_{42}\\ c_{Y2}=g_{12}g_{33}g_{44}-g_{12}g_{34}g_{43}-g_{13}g_{32}g_{44}+g_{13}g_{34}g_{42}+g_{14}g_{32}g_{43}-g_{14}g_{33}g_{42}\\ c_{Y3}=-g_{12}g_{23}g_{44}+g_{12}g_{24}g_{43}+g_{13}g_{22}g_{44}-g_{13}g_{24}g_{42}-g_{14}g_{22}g_{43}+g_{14}g_{23}g_{42}\\ c_{Y4}=g_{12}g_{23}g_{34}-g_{12}g_{24}g_{33}-g_{13}g_{22}g_{34}+g_{13}g_{24}g_{32}+g_{14}g_{22}g_{33}-g_{14}g_{23}g_{32}\\ c_{Y11}=-g_{21}g_{33}g_{44}+g_{21}g_{34}g_{43}+g_{23}g_{31}g_{44}-g_{23}g_{34}g_{41}-g_{24}g_{31}g_{43}+g_{24}g_{33}g_{41}\\ c_{Y22}=g_{11}g_{33}g_{44}-g_{11}g_{34}g_{43}-g_{13}g_{31}g_{44}+g_{13}g_{34}g_{41}+g_{14}g_{31}g_{43}-g_{14}g_{33}g_{41}\\ c_{Y33}=-g_{11}g_{23}g_{44}+g_{11}g_{24}g_{43}+g_{13}g_{21}g_{44}-g_{13}g_{24}g_{41}-g_{14}g_{21}g_{43}+g_{14}g_{23}g_{41}\\ c_{Y44}=g_{11}g_{23}g_{34}-g_{11}g_{24}g_{33}-g_{13}g_{21}g_{34}+g_{13}g_{24}g_{31}+g_{14}g_{21}g_{33}-g_{14}g_{23}g_{31}\\ c_{Z1}=-g_{21}g_{32}g_{44}+g_{21}g_{34}g_{42}+g_{22}g_{31}g_{44}-g_{22}g_{34}g_{41}-g_{24}g_{31}g_{42}+g_{24}g_{32}g_{41}\\ c_{Z2}=g_{11}g_{32}g_{44}-g_{11}g_{34}g_{42}-g_{12}g_{31}g_{44}+g_{12}g_{34}g_{41}+g_{14}g_{31}g_{42}-g_{14}g_{32}g_{41}\\ c_{Z3}=-g_{11}g_{22}g_{44}+g_{11}g_{24}g_{42}+g_{12}g_{21}g_{44}-g_{12}g_{24}g_{41}-g_{14}g_{21}g_{42}+g_{14}g_{22}g_{41}\\ c_{Z4}=g_{11}g_{22}g_{34}-g_{11}g_{24}g_{32}-g_{12}g_{21}g_{34}+g_{12}g_{24}g_{31}+g_{14}g_{21}g_{32}-g_{14}g_{22}g_{31}\\ c_{Z11}=-g_{21}g_{32}g_{43}+g_{21}g_{33}g_{42}+g_{22}g_{31}g_{43}-g_{22}g_{33}g_{41}-g_{23}g_{31}g_{42}+g_{23}g_{32}g_{41}\\ c_{Z22}=g_{11}g_{32}g_{43}-g_{11}g_{33}g_{42}-g_{12}g_{31}g_{43}+g_{12}g_{33}g_{41}+g_{13}g_{31}g_{42}-g_{13}g_{32}g_{41}\\ c_{Z33}=-g_{11}g_{22}g_{43}+g_{11}g_{23}g_{42}+g_{12}g_{21}g_{43}-g_{12}g_{23}g_{41}-g_{13}g_{21}g_{42}+g_{13}g_{22}g_{41}\\ c_{Z44}=g_{11}g_{22}g_{33}-g_{11}g_{23}g_{32}-g_{12}g_{21}g_{33}+g_{12}g_{23}g_{31}+g_{13}g_{21}g_{32}-g_{13}g_{22}g_{31} \end{array}\right.$$
(21)$$c_{X1}=\left[\begin{array}{cccc} \frac{3}{5} & -\frac{1}{20} & 0 & 0\\ -\frac{1}{20} & \frac{1}{15} & 0 & 0\\ 0 & 0 & \frac{1}{2}+\frac{1}{11\left(13h_{s}+1\right)} & -\frac{9}{2{\left(120h_{s}+17\right)}^{2}}\\ 0 & 0 & -\frac{1}{21{\left(13h_{s}+1\right)}^{2}} & \frac{1}{25}+\frac{1}{40\left(13h_{s}+1\right)} \end{array}\right]$$
(22)$$c_{X2}=\left[\begin{array}{cccc} -\frac{1}{700} & -\frac{1}{1400} & 0 & 0\\ -\frac{1}{1400} & -\frac{11}{6300} & 0 & 0\\ 0 & 0 & \frac{3}{8}-\frac{40h_{s}^{2}+6h_{s}+1}{{\left(13h_{s}+1\right)}^{3}} & \frac{1}{24}+\frac{h_{s}}{4}-\frac{120h_{s}^{2}+20h_{s}+1}{20{\left(13h_{s}+1\right)}^{3}}\\ 0 & 0 & \frac{1}{24}+\frac{h_{s}}{4}-\frac{120h_{s}^{2}+20h_{s}+1}{20{\left(13h_{s}+1\right)}^{3}} & \frac{1}{44}+\frac{h_{s}}{11}-\frac{84h_{s}^{2}+21h_{s}+1}{{\left(13h_{s}+1\right)}^{3}} \end{array}\right]$$
(23)$$c_{X3}=\left[\begin{array}{cccc} 0 & 0 & 0 & \frac{1}{120}\\ 0 & 0 & \frac{1}{120} & 0\\ 0 & \frac{1}{120} & 0 & 0\\ \frac{1}{120} & 0 & 0 & 0 \end{array}\right]$$
(24)$$c_{X11}=\left[\begin{array}{cccc} 0 & 0 & 0 & -\frac{1}{2}\\ 0 & 0 & -\frac{1}{2} & -\frac{1}{4}\\ 0 & -\frac{1}{2} & 0 & 0\\ -\frac{1}{2} & -\frac{1}{4} & 0 & 0 \end{array}\right]$$
(25)$$c_{X22}=\left[\begin{array}{cccc} \frac{1}{5} & -\frac{1}{10} & 0 & 0\\ -\frac{1}{10} & \frac{1}{20} & 0 & 0\\ 0 & 0 & \frac{1}{5} & \frac{1}{10}\\ 0 & 0 & \frac{1}{10} & \frac{1}{20} \end{array}\right]$$
(26)$$c_{X33}=\left[\begin{array}{cccc} 0 & 0 & 0 & \frac{1}{120}\\ 0 & 0 & \frac{1}{120} & 0\\ 0 & \frac{1}{120} & 0 & 0\\ \frac{1}{120} & 0 & 0 & 0 \end{array}\right]$$

The variables in the system expressed as Equation (20) are defined in Table 1.

## 3. Chain Model of the Spatial FLS

The chain model of a spatial FLS with large deformation is illustrated in Figure 3. The forces and torques loaded on the end of the FLS are *F*_xO_, *F*_yO_, *F*_zO_ and *M*_xO_, *M*_yO_, *M*_zO_, respectively, while the displacement and rotation angles of the FLS end are *p*_N_, *q*_N_, *r*_N_ and *θ*_xdN_, *θ*_yN_, *θ*_zN_, respectively. Subsequently, the FLS is divided into several flexure elements with equal length *L*_i_. Every flexure element has two endpoints and is modeled with the six-DOF compliance model introduced in Section 2. The load and deformation are defined in Figure 4. It is assumed that the FLS is divided into N flexure elements. The local coordinate system *x*_i_-*y*_i_-*z*_i_ is established at the endpoint (node i) of the element i and along its tangent direction. The local coordinate system of the first element is established at the fixed end of the FLS (node 0).

The free end of the FLS is defined as node N. The forces and torques loaded on the i-th flexure element are Δ*F*_xi_, Δ*F*_yi_, Δ*F*_zi_ and Δ*M*_xi_, Δ*M*_yi_, Δ*M*_zi_, respectively, while the corresponding displacements and rotation angles are Δ*p*_i_, Δ*q*_i_, Δ*r*_i_ and Δ*θ*_xdi_, Δ*θ*_yi_, Δ*θ*_zi_, respectively. The transformation matrix between the coordinate system on the end of the i-th flexure element and that on the endpoint (node i) is given as Equation (27):(27)$$\begin{array}{l} R_{\mathrm{di}}=R_{\mathrm{xi}}\left(\Delta \theta_{\mathrm{xdi}}\right)R_{\mathrm{yi}}\left(\Delta \theta_{\mathrm{yi}}\right)R_{\mathrm{zi}}\left(\Delta \theta_{\mathrm{zi}}\right)\\ =\left[\begin{array}{ccc} 1 & 0 & 0\\ 0 & c\left(\Delta \theta_{\mathrm{xdi}}\right) & -s\left(\Delta \theta_{\mathrm{xdi}}\right)\\ 0 & s\left(\Delta \theta_{\mathrm{xdi}}\right) & c\left(\Delta \theta_{\mathrm{xdi}}\right) \end{array}\right]\left[\begin{array}{ccc} c\left(\Delta \theta_{\mathrm{yi}}\right) & 0 & s\left(\Delta \theta_{\mathrm{yi}}\right)\\ 0 & 1 & 0\\ -s\left(\Delta \theta_{\mathrm{yi}}\right) & 0 & c\left(\Delta \theta_{\mathrm{yi}}\right) \end{array}\right]\left[\begin{array}{ccc} c\left(\Delta \theta_{\mathrm{zi}}\right) & -s\left(\Delta \theta_{\mathrm{zi}}\right) & 0\\ s\left(\Delta \theta_{\mathrm{zi}}\right) & c\left(\Delta \theta_{\mathrm{zi}}\right) & 0\\ 0 & 0 & 1 \end{array}\right]\\ =\left[\begin{array}{ccc} c\left(\Delta \theta_{\mathrm{yi}}\right)c\left(\Delta \theta_{\mathrm{zi}}\right) & c\left(\Delta \theta_{\mathrm{zi}}\right)s\left(\Delta \theta_{\mathrm{xdi}}\right)c\left(\Delta \theta_{\mathrm{yi}}\right)-c\left(\Delta \theta_{\mathrm{xdi}}\right)s\left(\Delta \theta_{\mathrm{zi}}\right) & s\left(\Delta \theta_{\mathrm{zi}}\right)s\left(\Delta \theta_{\mathrm{xdi}}\right)+c\left(\Delta \theta_{\mathrm{xdi}}\right)c\left(\Delta \theta_{\mathrm{zi}}\right)s\left(\Delta \theta_{\mathrm{yi}}\right)\\ c\left(\Delta \theta_{\mathrm{yi}}\right)s\left(\Delta \theta_{\mathrm{zi}}\right) & c\left(\Delta \theta_{\mathrm{zi}}\right)c\left(\Delta \theta_{\mathrm{xdi}}\right)+s\left(\Delta \theta_{\mathrm{xdi}}\right)s\left(\Delta \theta_{\mathrm{zi}}\right)s\left(\Delta \theta_{\mathrm{yi}}\right) & c\left(\Delta \theta_{\mathrm{xdi}}\right)s\left(\Delta \theta_{\mathrm{yi}}\right)s\left(\Delta \theta_{\mathrm{zi}}\right)-c\Delta \theta_{\mathrm{zi}}s\left(\Delta \theta_{\mathrm{xdi}}\right)\\ -s\left(\Delta \theta_{\mathrm{yi}}\right) & c\left(\Delta \theta_{\mathrm{yi}}\right)s\left(\Delta \theta_{\mathrm{xdi}}\right) & c\left(\Delta \theta_{\mathrm{xdi}}\right)c\left(\Delta \theta_{\mathrm{yi}}\right) \end{array}\right] \end{array}$$

The transformation matrix between the local coordinate system *x*_i_-*y*_i_-*z*_i_ of the i-th element and the global coordinate system *X*-*Y*-*Z* is as follows:(28)$$R_{i}=\overset{i}{\underset{m=1}{\Pi }}R_{\mathrm{dm}}$$
which can be defined as Equation (29):(29)$$\left[\begin{array}{c} T_{x}\\ T_{y}\\ T_{z} \end{array}\right]=R_{i}^{-1}\left[\begin{array}{c} 1\\ 0\\ 0 \end{array}\right]$$

The value of every angle can be calculated using the above transformation matrix, and the calculation process is given as Equation (30):(30)$$\{\begin{array}{l} \theta_{\mathrm{yi}}=\arctan\frac{-T_{z}}{T_{x}}\\ \theta_{\mathrm{zi}}=\arctan\frac{T_{y}}{T_{x}}\; \end{array}$$
since
(31)$$R_{i}=R_{\mathrm{xi}}\left(\theta_{\mathrm{xdi}}\right)R_{\mathrm{yi}}\left(\theta_{\mathrm{yi}}\right)R_{\mathrm{zi}}\left(\theta_{\mathrm{zi}}\right)$$

Equation (32) can be deduced:(32)$$R_{\mathrm{xi}}\left(\theta_{\mathrm{xdi}}\right)=R_{i}R_{\mathrm{yi}}^{-1}\left(\theta_{\mathrm{yi}}\right)R_{\mathrm{zi}}^{-1}\left(\theta_{\mathrm{zi}}\right)$$

By further transforming Equations (32) and (33) can be derived as follows:(33)$$\theta_{\mathrm{xdi}}=\arctan\frac{\sin\theta_{\mathrm{xdi}}}{\cos\theta_{\mathrm{xdi}}}$$

The total displacement of node i in the chain model can be expressed as follows:(34)$$\left[\begin{array}{c} p_{i}\\ q_{i}\\ r_{i} \end{array}\right]=\left[\begin{array}{c} p_{1}\\ q_{1}\\ r_{1} \end{array}\right]+\sum_{j=2}^{i}\left(\overset{j-1}{\underset{m=1}{\Pi }}R_{m}^{-1}\left[\begin{array}{c} \Delta p_{i}\\ \Delta q_{i}\\ \Delta r_{i} \end{array}\right]\right)$$
where
(35)$$\left[\begin{array}{c} \Delta p_{i}\\ \Delta q_{i}\\ \Delta r_{i} \end{array}\right]=\left[\begin{array}{c} \Delta U_{\mathrm{Xi}}+L/N\\ \Delta U_{\mathrm{Yi}}\\ \Delta U_{\mathrm{Zi}} \end{array}\right]$$

The loads applied on node i are:(36)$$\{\begin{array}{l} F_{\mathrm{xi}}=\Delta F_{\mathrm{xi}}=F_{\mathrm{xO}}\\ F_{\mathrm{yi}}=\Delta F_{\mathrm{yi}}=F_{\mathrm{yO}}\\ F_{\mathrm{zi}}=\Delta F_{\mathrm{zi}}=F_{\mathrm{zO}}\\ M_{\mathrm{xi}}=M_{\mathrm{xO}}+\left(q_{N}-q_{i}\right)F_{\mathrm{zO}}-\left(r_{N}-r_{i}\right)F_{\mathrm{yO}}\\ M_{\mathrm{yi}}=M_{\mathrm{yO}}+\left(p_{N}-p_{i}\right)F_{\mathrm{zO}}+\left(r_{N}-r_{i}\right)F_{\mathrm{xO}}\;\\ M_{\mathrm{zi}}=M_{\mathrm{zO}}+\left(p_{M}-p_{i}\right)F_{\mathrm{yO}}-\left(q_{N}-q_{i}\right)F_{\mathrm{xO}} \end{array}$$

The transformation relationship between the loads in the local coordinate system *x*_i_-*y*_i_-*z*_i_ and those in the global coordinate system *X*-*Y*-*Z* is:(37)$$\left[\begin{array}{c} \Delta M_{\mathrm{xi}}\\ \Delta M_{\mathrm{yi}}\\ \Delta M_{\mathrm{zi}} \end{array}\right]=\left(\overset{I-1}{\underset{e=1}{\Pi }}R_{i-e}\right)\left[\begin{array}{c} M_{\mathrm{xi}}\\ M_{\mathrm{yi}}\\ M_{\mathrm{zi}} \end{array}\right]$$
(38)$$\left[\begin{array}{c} \Delta F_{\mathrm{xi}}\\ \Delta F_{\mathrm{yi}}\\ \Delta F_{\mathrm{zi}} \end{array}\right]=\left(\overset{j-1}{\underset{e=1}{\Pi }}R_{i-e}\right)\left[\begin{array}{c} F_{1}\\ F_{1}\\ F_{1} \end{array}\right]=\left(\overset{j-1}{\underset{e=1}{\Pi }}R_{i-e}\right)\left[\begin{array}{c} F_{\mathrm{xO}}\\ F_{\mathrm{yO}}\\ F_{\mathrm{zO}} \end{array}\right]$$

Furthermore, the relationship between the forces exerted on adjacent nodes is:(39)$$\left[\begin{array}{c} \Delta F_{x\left(i+1\right)}\\ \Delta F_{y\left(i+1\right)}\\ \Delta F_{z\left(i+1\right)} \end{array}\right]=R_{i}\left[\begin{array}{c} \Delta F_{\mathrm{xi}}\\ \Delta F_{\mathrm{yi}}\\ \Delta F_{\mathrm{zi}} \end{array}\right]=R_{i}R_{i-1}\left[\begin{array}{c} \Delta F_{x\left(i-1\right)}\\ \Delta F_{y\left(i-1\right)}\\ \Delta F_{z\left(i-1\right)} \end{array}\right]$$

The chain model of the spatial FLS can be obtained by combining the equation systems of every flexure element and using the six-DOF compliance model introduced in Section 2 to model every flexure element. Subsequently, a relationship between loads (*F*_xO_, *F*_yO_, and *F*_zO_), torques (*M*_xO_, *M*_yO_, and *M*_zO_), displacements (*p*_N_, *q*_N_, and *r*_N_), and rotation angles (*θ*xdN, *θ*yN, and *θ*zN) can be obtained. Due to the complexity of the equation system, Newton’s method is used to solve it. The calculation results (*p*_N_, *q*_N_, *r*_N_ and *θ*_xdN_, *θ*_yN_, *θ*_zN_) gradually approach a fixed value with an increasing number of flexure elements n. Consequently, the chain model can be considered as convergent.

## 4. Parametric Analysis

### 4.1. Compliance

According to the previous sections, the relationship between deformation and load of the spatial FLS under large deformation is nonlinear. In this section, the effects of the structural parameters on the compliance of the FLS as a function of the deformation along the thickness direction (working direction) are analyzed. Based on the variables in the chain model introduced in Section 2.3, the compliance in every direction can be defined as follows:(40)$$\{c_{x}=\frac{p_{N}}{F_{\mathrm{xO}}},c_{y}=\frac{q_{N}}{F_{\mathrm{yO}}},c_{z}=\frac{r_{N}}{F_{\mathrm{zO}}},c_{\mathrm{rx}}=\frac{\theta_{\mathrm{xdN}}}{M_{\mathrm{xO}}}$$

The range of each parameter is as follows:The range of *w* is [80 mm, 120 mm];The range of *L* is [200 mm, 250 mm];The range of *t* is [0.6 mm, 1 mm].

The material of the FLS was set as spring steel (60Si2Mn) with a Young’s modulus of *E* = 206 GPa and Poisson’s ratio of *v* = 0.29. The effects of the different parameters are presented in Figure 5, Figure 6, Figure 7 and Figure 8, where (a–c) show the effects of *L*, *w*, and *t*, respectively, on compliance.

In Figure 5, it can be observed that the value of *c*_y_ decreased with the increase of the FLS deformation along the thickness direction. Moreover, the value of *c*_y_ increased with the increase of *L*, decreased with the increase of *w*, and increased with the increase of *t*. The variation range of *c*_y_ increased with increasing *L*, decreased with increasing *w*, and decreased with increasing *t* over the deformation range of the FLS.

As can be seen in Figure 6, the value of *c*_z_ increased with the increase of the FLS deformation along the thickness direction. The value of *c*_z_ increased with increasing *L*, decreased with increasing *w*, and increased with increasing *t*. The variation range of *c*_z_ increased with increasing *L*, decreased with increasing *w*, and decreased with increasing *t* over the deformation range of the FLS.

According to Figure 7, the value of *c*_x_ increased with the increase of the FLS deformation along the thickness direction. The value of *c*_x_ increased with increasing *L*, decreased with increasing *w*, and decreased with increasing *t*. The variation range of *c*_x_ increased with increasing *L*, decreased with increasing *w*, and decreased with increasing *t* over the deformation range of the FLS.

In Figure 8, it can be seen that the value of *c*_rx_ decreased with the increase of the FLS deformation along the thickness direction. More specifically, the value of *c*_rx_ increased with increasing *L*, decreased with increasing *w*, and decreased with increasing *t*. The variation range of *c*_rx_ did not change with the variation of *L* and that of *t* but decreased with increasing *w* over the deformation range of the FLS.

### 4.2. Compliance Ratio

The ratio of the compliance along the working direction to that along the non-working directions is an important indicator that can reflect the motion stability of the FLS. The compliance ratio indicates whether the performance of the FLS can ensure the smoothness of motion in the working direction and resist the disturbance forces in the non-working directions. To this end, the following compliance ratios were analyzed:(41)$$\{\frac{c_{x}}{c_{y}},\frac{c_{z}}{c_{y}},\frac{c_{\mathrm{rx}}}{c_{y}}$$

The range of each parameter is as follows:The range of *w* is [80 mm, 120 mm];The range of *L* is [200 mm, 250 mm];The range of *t* is [0.6 mm, 1 mm].

The material of the FLS was set as spring steel (60Si2Mn) with a Young’s modulus of *E* = 206 GPa and Poisson’s ratio of *v* = 0.29. The effects of the different parameters are exhibited in Figure 9, Figure 10 and Figure 11, where (a–c) show the effects of *L*, *w*, and *t*, respectively, on compliance ratio.

As can be seen in Figure 9, the value of *c*_z_/*c*_y_ increased with the increase of the FLS deformation along the thickness direction. The value of *c*_z_/*c*_y_ did not change with the variation of *L*, decreased with increasing *w* and increased with increasing *t*. The variation range of *c*_z_/*c*_y_ did not change with the variation of *L*, decreased with increasing *w*, and increased with increasing *t* over the deformation range of the FLS.

In Figure 10, it can be observed that the value of *c*_x_/*c*_y_ increased with the increase of the FLS deformation along the thickness direction. The value of *c*_x_/*c*_y_ decreased with increasing *L*, remained unchanged with the variation of *w*, and increased with increasing *t*. The variation range of *c*_x_/*c*_y_ decreased with increasing *L*, did not change with the variation of *w*, and increased with increasing *t* over the deformation range of the FLS.

According to Figure 11, the value of *c*_rx_/*c*_y_ decreased first and then increased with the increase of the FLS deformation along the thickness direction. The value of *c*_rx_/*c*_y_ decreased with increasing *L*, remained unchanged with the variation of *w*, and increased with increasing *t*. The variation range of *c*_rx_/*c*_y_ decreased with increasing *L*, did not change with the variation of *w*, and increased with increasing *t* over the deformation range of the FLS.

## 5. Theoretical Model Verification

In this paper, the compliance model of the spatial large deformation FLS is verified both experimentally and through finite element simulations. The structure of the FLS, the nodes of the chain model, and the positions of the applied load are depicted in Figure 12. The structural parameters of the FLS are given in Table 2. The chain model described in Section 3 was used to discretize the FLS. More specifically, the FLS was divided into 18 flexure elements; the length of each element was 10 mm. The holes for applying different loads to the FLS were placed on the boundary where node 16 was located. The distance between adjacent holes was 20 mm. In both the simulation and experiment, the deformation and rotation angle of the midpoints (*P*_1_, *P*_2_, *P*_3_) on the discrete boundary where nodes 3, 8, and 13 are located were compared with the calculation results obtained by the theoretical model. In order to analyze the rotation angle of the FLS in the simulation and experimental results, the deformations of the points (*P*_11_, *P*_12_, *P*_21_, *P*_22_, *P*_31_, *P*_32_) on the boundary line on the FLS where nodes 3, 8, and 13 are located were extracted. The distance between the different points is given in Table 3. The coordinates of the points when the FLS was not deformed are listed in Table 4. The coordinates of each point on the FLS after deformation are listed in Table 5 and are expressed based on the chain model introduced in Section 3.

### 5.1. Finite Element Simulation

The mesh of the FLS model is depicted in Figure 13. The model was meshed with tetrahedral elements, the number of nodes was 228,711 and the number of elements was 111,623. In this paper, the deformation of the FLS under loads applied at three different positions was simulated. The applied load was 1 N, and the application positions of the loads are given in Table 6. The simulation results are demonstrated in Figure 14, and the points *P*_11_, *P*_12_, *P*_21_, *P*_22_, *P*_31_, *P*_32_ were recorded from the simulation results. The displacement and rotation angle of each point (*P*_1_, *P*_2_, *P*_3_) could be calculated based on the simulation results of points (*P*_11_, *P*_12_, *P*_21_, *P*_22_, *P*_31_, *P*_32_). The theoretical calculation and simulation results of points (*P*_1_, *P*_2_, *P*_3_) are listed in Table 7. It can be found that the error was smaller than 5%. This verifies that the theoretical model proposed in this paper is accurate.

### 5.2. Experiment

The experimental platform used to test the FLS is depicted in Figure 15. A 6-DOF manipulator (UNIVERSAL ROBOTS, UR5) carrying a surface structured light sensor (TECHLEGO, Q3; measurement accuracy of ±0.005 mm) was used to scan the deformed FLS and capture its spatial contours. The spatial change of the position of points (*P*_11_, *P*_12_, *P*_21_, *P*_22_, *P*_31_, *P*_32_) on the FLS after deformation was obtained by processing the spatial contour data. Subsequently, the deformation and rotation angle of points (*P*_1_, *P*_2_, *P*_3_) on the FLS were calculated. The FLS used in the experiment is depicted in Figure 16. The FLS was marked at points (*P*_11_, *P*_12_, *P*_21_, *P*_22_, *P*_31_, *P*_32_) in order to facilitate the subsequent data processing. Due to that the scanning area of the structured light sensor was small and could not fully cover the FLS, the FLS needed to be marked at reference points to implement the splicing algorithm of spatial contour. Then, the scanned contour data of the entire FLS could be obtained. The FLS deformed under load is displayed in Figure 17. The load was *m* = 100 g and the gravitational acceleration *g* = 9.8066 N/kg. The spatial contours of the deformed FLS are presented in Figure 18. The experimental results of the points (*P*_1_, *P*_2_, *P*_3_) were obtained after processing the spatial contour data and are listed in Table 8. The error was smaller than 5%. Consequently, the accuracy of the theoretical model proposed in this paper was again verified.

## 6. Conclusions

In this paper, a 6-DOF compliance model for FLSs with large *w*/*L* under large deformation has been proposed. A new differential equation system is established based on Timoshenko beam theory along the width direction. The shear deformation along the width direction of the FLS with large *w*/*L* and the spatial geometric nonlinearity can be accurately described by this method. In Section 2, a compliance model for flexure elements with small deformation (i.e., <0.1 *L*) has been proposed. Then, the chain model of the spatial FLS was established in Section 3. The flexure elements with small deformation are assembled and expanded to accurately represent the large deformation of the FLS. Based on the chain model and the compliance model of the flexure elements, the effects of the structural parameters on the compliance and the compliance ratio of the FLS under large deformation were analyzed in Section 4. In Section 5, the theoretical model was verified experimentally and through finite element simulations, and the accuracy of the proposed theoretical model was proven. It was found that the relative error between theoretical result and experiment result was less than 5%.

## Figures and Tables

**Figure 1 micromachines-13-01090-f001:**
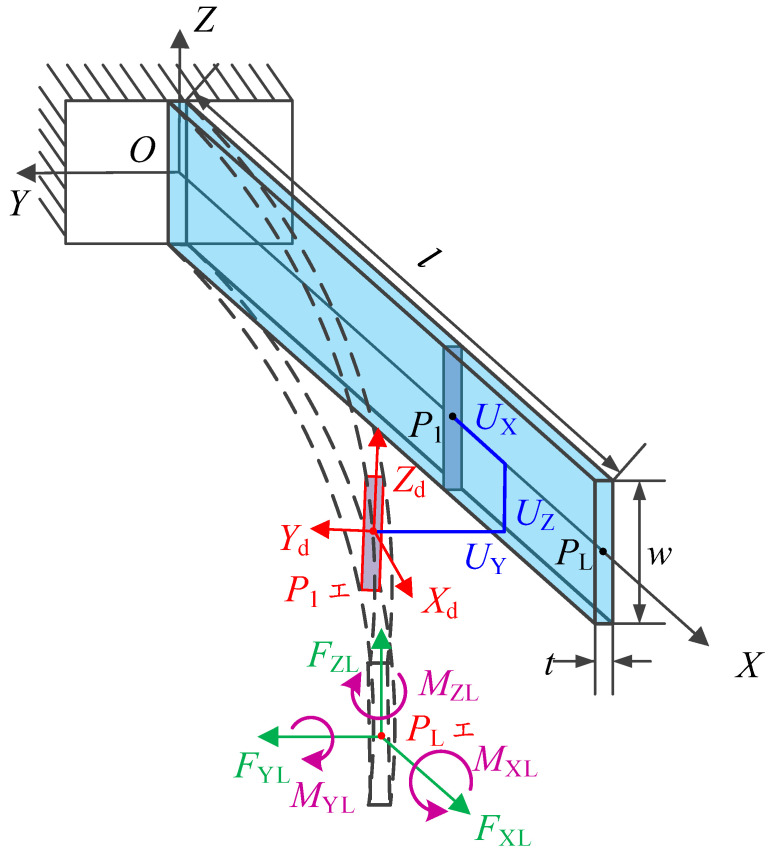
Six-DOF spatial deformation of the FLS.

**Figure 2 micromachines-13-01090-f002:**
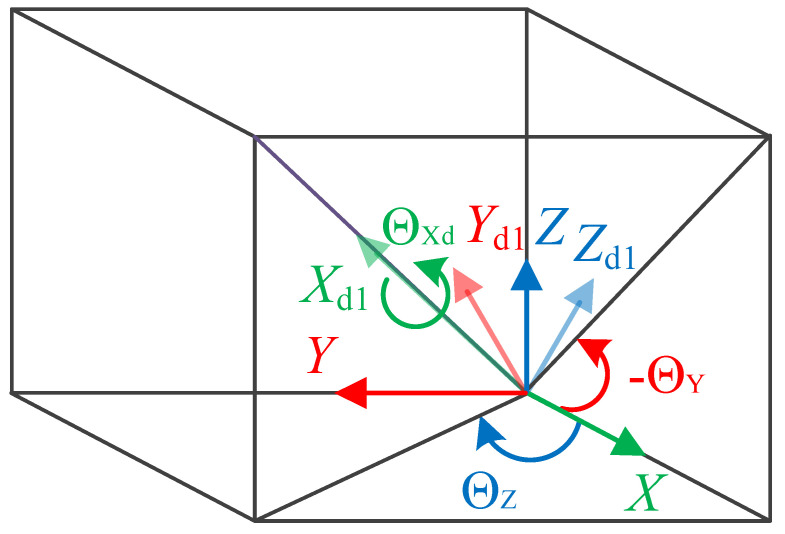
Transformation of the spatial coordinate systems.

**Figure 3 micromachines-13-01090-f003:**
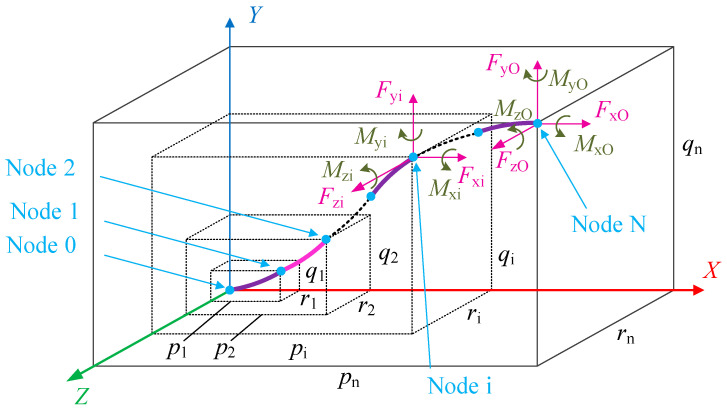
Chain model of the spatial FLS.

**Figure 4 micromachines-13-01090-f004:**
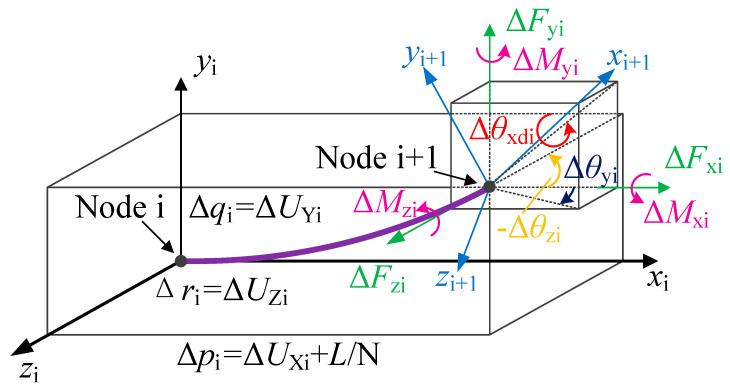
Loads and coordinate transformation of a spatial flexure element.

**Figure 5 micromachines-13-01090-f005:**
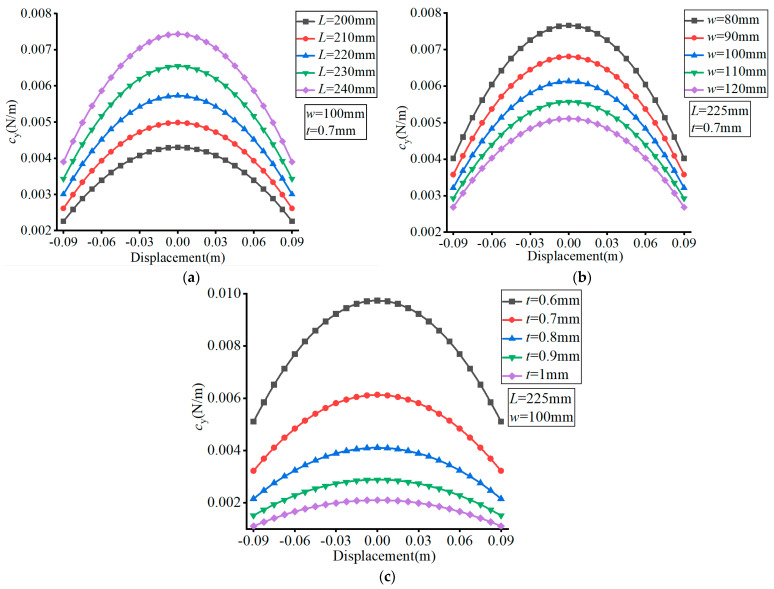
Effect of structural parameters on *c*_y_. (**a**) Effect of *L*; (**b**) Effect of *w*; (**c**) Effect of *t*.

**Figure 6 micromachines-13-01090-f006:**
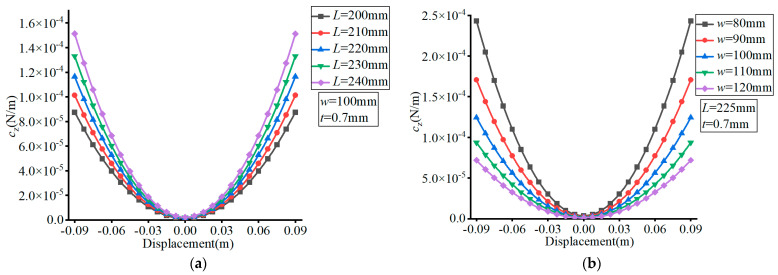
Effect of structural parameters on *c*_z_. (**a**) Effect of *L*; (**b**) Effect of *w*; (**c**) Effect of *t*.

**Figure 7 micromachines-13-01090-f007:**
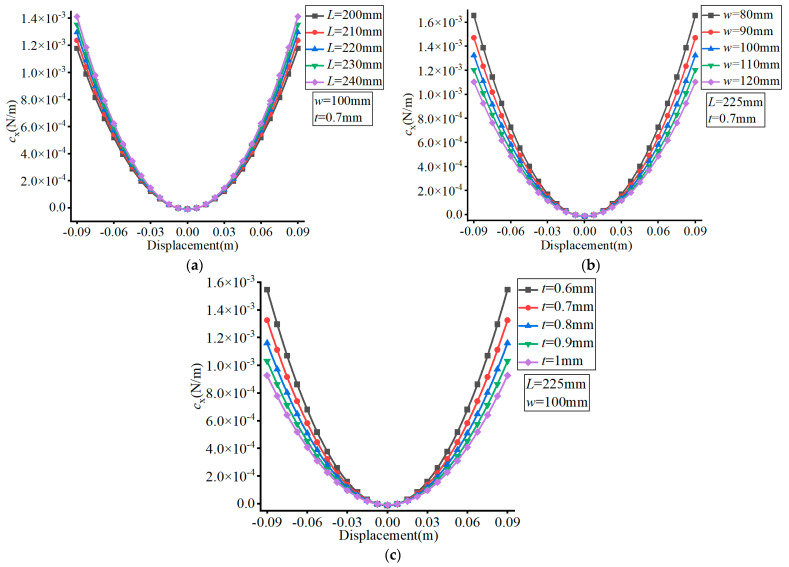
Effect of structural parameters on *c*_x_. (**a**) Effect of *L*; (**b**) Effect of *w*; (**c**) Effect of *t*.

**Figure 8 micromachines-13-01090-f008:**
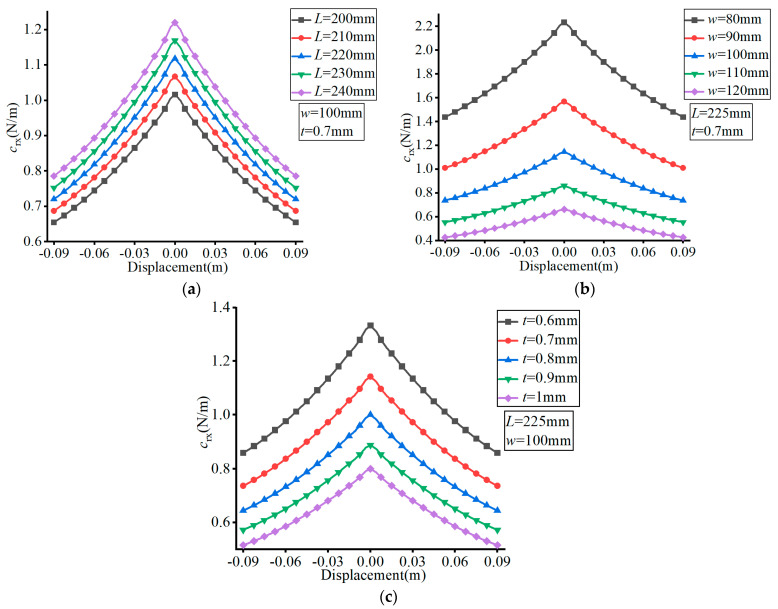
Effect of structural parameters on *c*_rx_. (**a**) Effect of *L*; (**b**) Effect of *w*; (**c**) Effect of *t*.

**Figure 9 micromachines-13-01090-f009:**
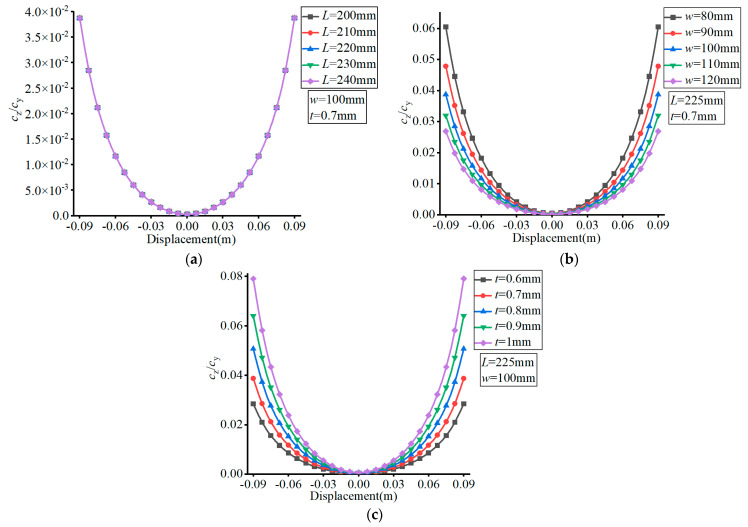
Effect of structural parameters on *c*_z_/*c*_y_. (**a**) Effect of *L*; (**b**) Effect of *w*; (**c**) Effect of *t*.

**Figure 10 micromachines-13-01090-f010:**
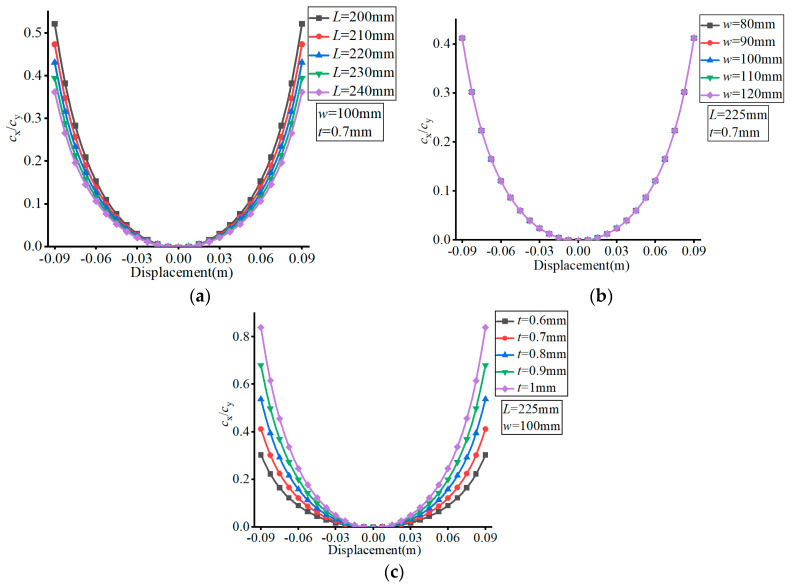
Effect of structural parameters on *c*_x_/*c*_y_. (**a**) Effect of *L*; (**b**) Effect of *w*; (**c**) Effect of *t*.

**Figure 11 micromachines-13-01090-f011:**
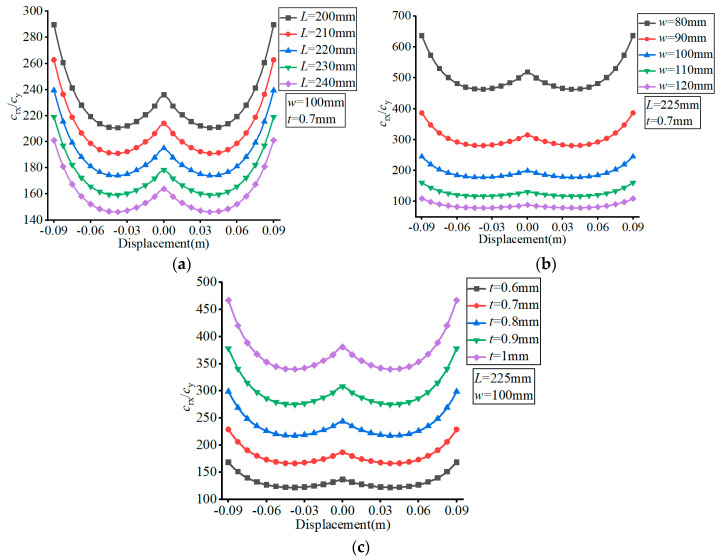
Effect of structural parameters on *c*_rx_/*c*_y_. (**a**) Effect of *L*; (**b**) Effect of *w*; (**c**) Effect of *t*.

**Figure 12 micromachines-13-01090-f012:**
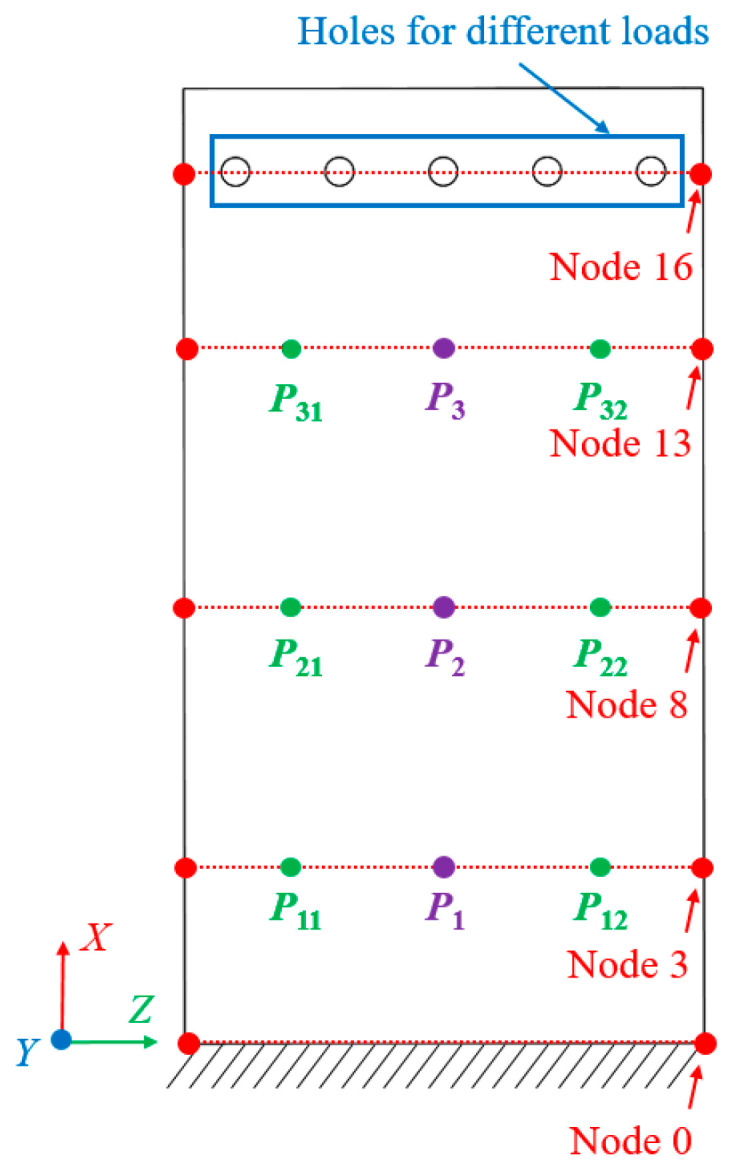
Nodes in the chain model (*P*_1_, *P*_2_, *P*_3_) and nodes used for displacement analysis (*P*_11_, *P*_12_, *P*_21_, *P*_22_, *P*_31_, *P*_32_).

**Figure 13 micromachines-13-01090-f013:**
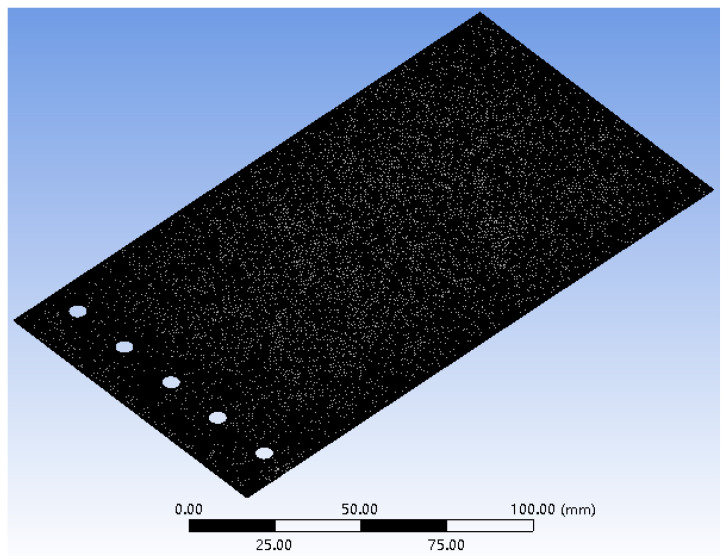
The finite element mesh of the FLS model.

**Figure 14 micromachines-13-01090-f014:**
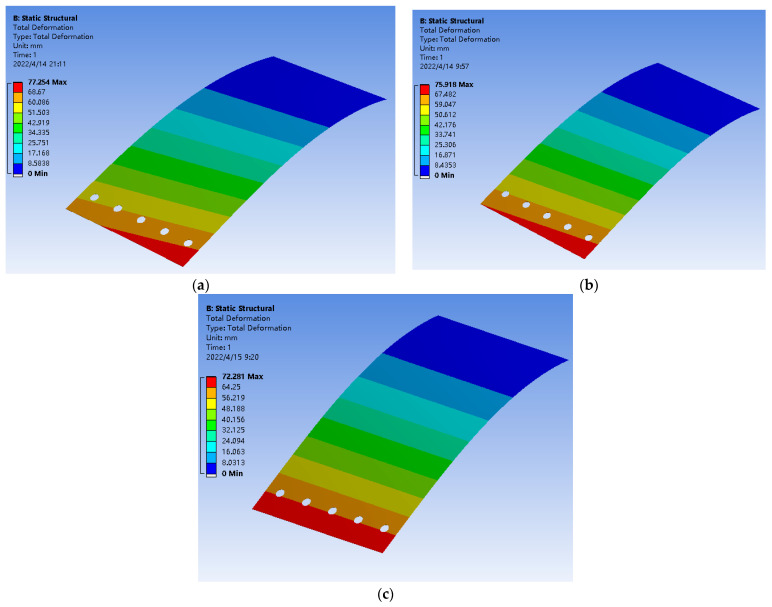
Finite element simulation results of the FLS under different loading conditions. (**a**) Case 1; (**b**) Case 2; (**c**) Case 3.

**Figure 15 micromachines-13-01090-f015:**
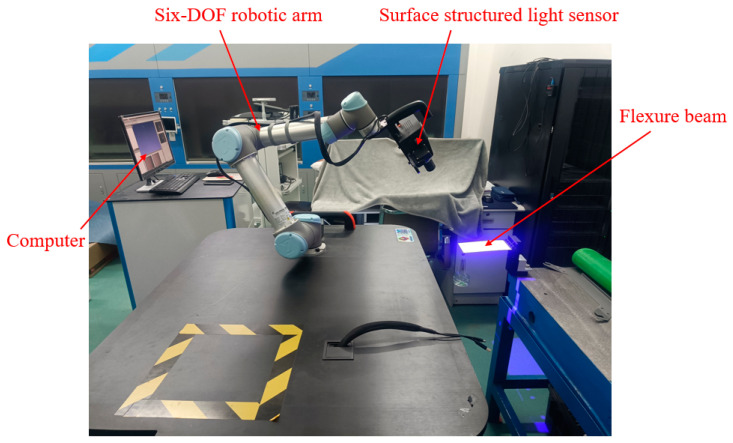
Experimental platform used for the compliance testing of the FLS.

**Figure 16 micromachines-13-01090-f016:**
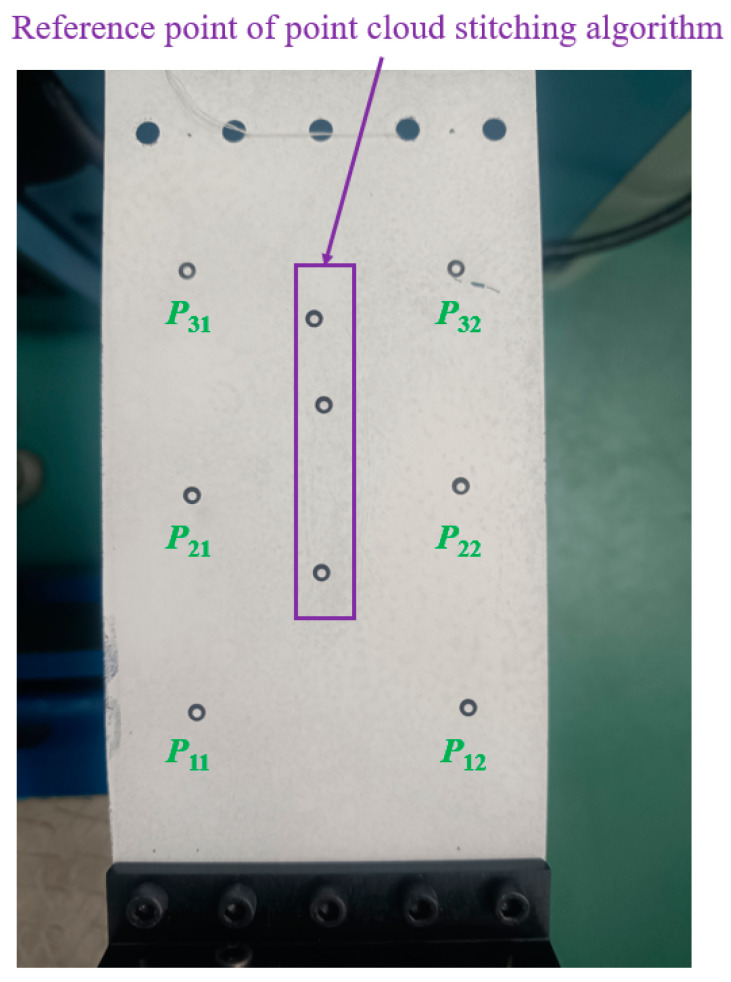
The FLS in the experiment.

**Figure 17 micromachines-13-01090-f017:**
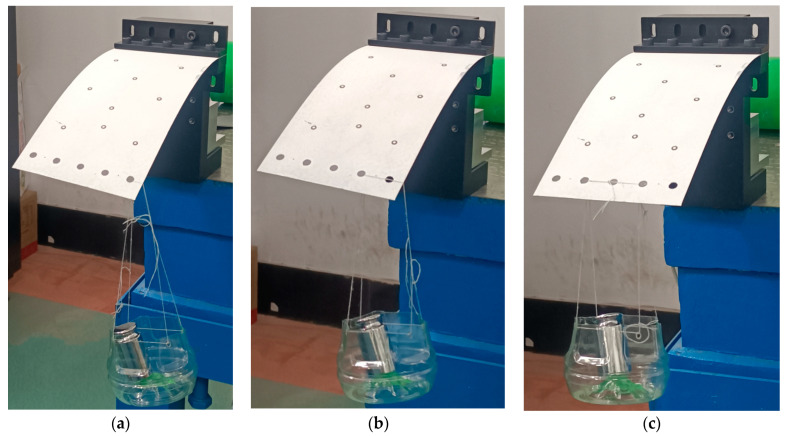
Deformation of the FLS under the different loading cases. (**a**) Case 1; (**b**) Case 2; (**c**) Case 3.

**Figure 18 micromachines-13-01090-f018:**
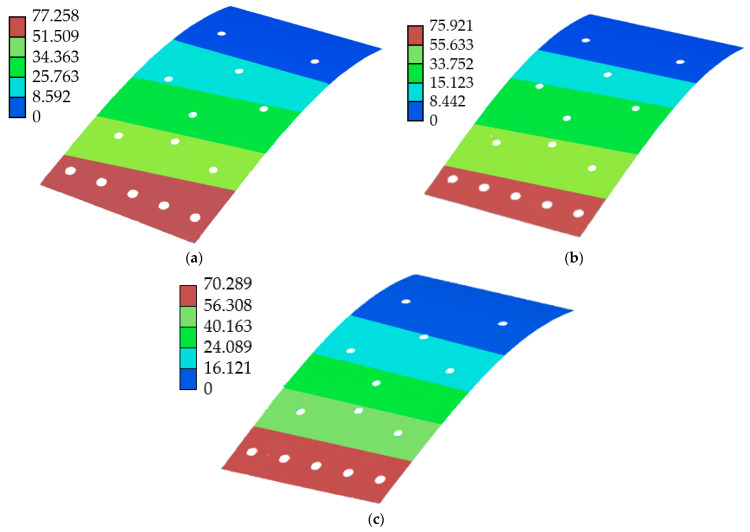
Spatial contours of the deformed FLS under the different loading cases. (**a**) Case 1; (**b**) Case 2; (**c**) Case 3.

**Table 1 micromachines-13-01090-t001:** Expressions of the variables in Equation (20).

Variable	Expression
g_11_	$-\frac{1}{700}f_{X1}^{2}-\frac{1}{5}m_{X1}^{2}+\frac{6}{5}f_{X1}+12$
g_12_	$\frac{1}{1400}f_{X1}^{2}+\frac{1}{10}m_{X1}^{2}-\frac{1}{10}f_{X1}-6$
g_13_	0
g_14_	$-\frac{1}{60}f_{X1}m_{X1}+m_{X1}$
g_21_	$\frac{1}{1400}f_{X1}^{2}+\frac{1}{10}m_{X1}^{2}-\frac{1}{10}f_{X1}-6$
g_22_	$\frac{11}{6300}f_{X1}^{2}-\frac{1}{20}m_{X1}^{2}+\frac{2}{15}f_{X1}+4$
g_23_	$-\frac{1}{60}f_{X1}m_{X1}+m_{X1}$
g_24_	$-\frac{m_{X1}}{2}$
g_31_	$\frac{\left(1820h_{s}^{2}+192h_{s}+13\right)f_{X1}^{2}}{720\alpha {\left(11h_{s}+1\right)}^{3}}+\frac{\left(107h_{s}+13\right)f_{X1}}{123h_{s}+13}-\frac{m_{X0}^{2}}{5\alpha }+\frac{153}{120h_{s}+15}$
g_32_	$\frac{\left(1769h_{s}^{2}+177h_{s}+1\right)f_{X1}^{2}}{1293\alpha {\left(14h_{s}+1\right)}^{3}}+\left(\frac{203}{400}-\frac{639h_{s}+73}{5\left(120h_{s}+11\right)}\right)f_{X1}-\frac{m_{X0}^{2}}{10\alpha }-\frac{89}{120h_{s}+17}$
g_33_	0
g_34_	$-m_{X1}+\frac{f_{X1}m_{X1}}{60\alpha }$
g_41_	$\frac{\left(50h_{s}+1\right)f_{X1}^{2}}{1440\alpha {\left(11h_{s}+1\right)}^{3}}-\frac{141f_{X1}}{13{\left(123h_{s}+13\right)}^{2}}-\frac{m_{X0}^{2}}{10\alpha }-\frac{89}{120h_{s}+17}$
g_42_	$\frac{\left(1769h_{s}^{2}+177h_{s}+1\right)f_{X1}^{2}}{1293\alpha {\left(14h_{s}+1\right)}^{3}}+\left(\frac{203}{400}-\frac{639h_{s}+73}{5\left(120h_{s}+11\right)}\right)f_{X1}-\frac{m_{X0}^{2}}{10\alpha }-\frac{89}{120h_{s}+17}$
g_43_	0
g_44_	$-m_{X1}+\frac{f_{X1}m_{X1}}{60\alpha }$

**Table 2 micromachines-13-01090-t002:** Structural parameters.

Parameter	*L* (m)	*W* (m)	*T* (m)	Length of the Single Element (m)
Value	0.18	0.1	0.00023	0.01

**Table 3 micromachines-13-01090-t003:** The distance between points.

Points		Distance (m)
*P* _31_	*P* _3_	0.03
*P* _32_	*P* _3_	0.03
*P* _21_	*P* _2_	0.03
*P* _22_	*P* _2_	0.03
*P* _11_	*P* _1_	0.03
*P* _12_	*P* _1_	0.03
*P* _31_	*P* _3_	0.03

**Table 4 micromachines-13-01090-t004:** Coordinates of points when the FLS is not deformed.

Point	Coordinate (Unit: m)
*P* _1_	(0.03, 0, 0)
*P* _11_	(0.03, 0, −0.03)
*P* _12_	(0.03, 0, 0.03)
*P* _2_	(0.08, 0, 0.)
*P* _21_	(0.08, 0, −0.03)
*P* _22_	(0.08, 0, 0.03)
*P* _3_	(0.13, 0, 0.)
*P* _31_	(0.13, 0, −0.03)
*P* _32_	(0.13, 0, 0.03)

**Table 5 micromachines-13-01090-t005:** Coordinates of points after deformation.

Point	Coordinate (Unit: m)
*P* _1_	(0.03 + *p*_3_, *q*_3_, *r*_3_)
*P* _11_	(0.03 + *p*_3_ − 0.03 sin *θ*_y3_, *q*_3_ − 0.03 sin *θ*_xd3_, *r*_3_ − 0.06 + (1 − 0.03 cos *θ*_y3_ − 0.03 cos *θ*_xd3_))
*P* _12_	(0.03 + *p*_3_ + 0.03 sin *θ*_y3_, *q*_3_ + 0.03 sin *θ*_xd3_, *r*_3_ + 0.06 − (1 − 0.03 cos *θ*_y3_ − 0.03 cos *θ*_xd3_))
*P* _2_	(0.08 + *p*_8_, *q*_8_, *r*_8_)
*P* _21_	(0.08 + *p*_8_ − 0.03 sin *θ*_y__8_, *q*_8_ − 0.03 sin *θ*_xd__8_, *r*_8_ − 0.06 + (1 − 0.03 cos *θ*_y__8_ − 0.03 cos *θ*_xd__8_))
*P* _22_	(0.08 + *p*_8_ + 0.03 sin *θ*_y__8_, *q*_8_ + 0.03 sin *θ*_xd__8_, *r*_8_ + 0.06 − (1 − 0.03 cos *θ*_y__8_ − 0.03 cos *θ*_xd__8_))
*P* _3_	(0.13 + *p*_13_, *q*_13_, *r*_13_)
*P* _31_	(0.13 + *p*_13_ − 0.03 sin *θ*_y__13_, *q*_13_ − 0.03 sin *θ*_xd__13_, *r*_13_ − 0.06 + (1 − 0.03 cos *θ*_y__13_ − 0.03 cos *θ*_xd__13_))
*P* _32_	(0.13 + *p*_13_ + 0.03 sin *θ*_y__13_, *q*_13_ + 0.03 sin *θ*_xd__13_, *r*_3_ + 0.06 − (1 − 0.03 cos *θ*_y__13_ − 0.03 cos *θ*_xd__13_))

**Table 6 micromachines-13-01090-t006:** Coordinates of points after deformation.

Case	Position of Load	Load
1	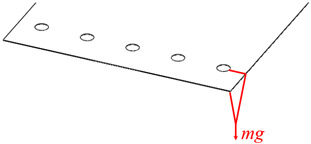	*M*_xO_ = 0.045 N × m
2	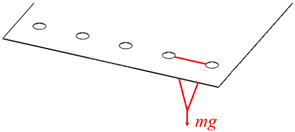	*M*_xO_ = 0.03 N × m
3	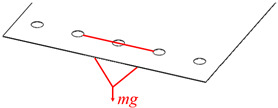	*F*_yO_ = −1 N

Remark: *mg* = 1 N.

**Table 7 micromachines-13-01090-t007:** Theoretical and simulation results of the displacement and rotation angle of points (*P*_1_, *P*_2_, *P*_3_).

Result	Case 1			Case 2			Case 3		
	Theoretical	Simulation	Error (%)	Theoretical	Simulation	Error (%)	Theoretical	Simulation	Error (%)
*p* _3_	−0.00023	−0.000231	0.43	−0.00024	−0.000239	0.42	−0.00021	−0.000208	0.95
*q* _3_	−0.00348	−0.003479	0.29	−0.00339	−0.003421	0.88	−0.00349	−0.00361	3.44
*r* _3_	0.00002	0.000021	5.00	0.00002	0.000019	5.00	0.00002	0.000019	5.00
*θ* _y3_	0.09237	0.092373	0.03	0.09357	0.09500	1.60	0	0.0001	-
*θ* _xd3_	0.87352	0.87431	0.09	0.67352	0.65900	2.16	0	0.0002	-
*p* _8_	−0.00253	−0.00254	0.39	−0.00278	−0.002792	0.43	−0.00284	−0.00277	2.46
*q* _8_	−0.01899	−0.018973	0.68	−0.01902	−0.018821	1.05	−0.01901	−0.01923	1.16
*r* _8_	0.00024	0.000247	2.92	0.00014	0.000141	0.71	0.00008	0.000083	3.75
*θ* _y8_	0.87000	0.87636	0.73	0.62300	0.61856	0.80	0	0.0001	-
*θ* _xd8_	2.97035	2.97128	0.31	2.23821	2.19700	1.85	0	0.0002	-
*p* _13_	−0.00891	−0.00879	1.35	−0.00897	−0.008911	0.67	−0.00900	−0.00919	2.11
*q* _13_	−0.04407	−0.04418	0.25	−0.04197	−0.042464	1.19	−0.04328	−0.04353	0.58
*r* _13_	0.00079	0.00080	1.27	0.00065	0.000659	1.38	0.00058	0.00059	1.72
*θ* _y13_	1.83254	1.84138	0.55	1.45724	1.447243	0.69	0	0.0003	-
*θ* _xd13_	4.56376	4.60134	0.82	3.62178	3.641776	0.55	0	0.0002	-

Remark: The unit of deformation is m and that of the rotation angle is °.

**Table 8 micromachines-13-01090-t008:** Theoretical and experimental results of the displacement and rotation angle of points (*P*_1_, *P*_2_, *P*_3_).

Result	Case 1			Case 2			Case 3		
	Theoretical	Experiment	Error (%)	Theoretical	Experiment	Error (%)	Theoretical	Experiment	Error (%)
*p* _3_	−0.00023	−0.000231	0.43	−0.00024	−0.000239	0.42	−0.00021	−0.000208	0.95
*q* _3_	−0.00348	−0.003479	0.29	−0.00339	−0.003421	0.88	−0.00349	−0.00361	3.44
*r* _3_	0.00002	0.000021	5.00	0.00002	0.000019	5.00	0.00002	0.000019	5.00
*θ* _y3_	0.09237	0.092373	0.03	0.09357	0.09500	1.60	0	0.0001	-
*θ* _xd3_	0.87352	0.87431	0.09	0.67352	0.65900	2.16	0	0.0002	-
*p* _8_	−0.00253	−0.00254	0.39	−0.00278	−0.002792	0.43	−0.00284	−0.00277	2.46
*q* _8_	−0.01899	−0.018973	0.68	−0.01902	−0.018821	1.05	−0.01901	−0.01923	1.16
*r* _8_	0.00024	0.000247	2.92	0.00014	0.000141	0.71	0.00008	0.000083	3.75
*θ* _y8_	0.87000	0.87636	0.73	0.62300	0.61856	0.80	0	0.0001	-
*θ* _xd8_	2.97035	2.97128	0.31	2.23821	2.19700	1.85	0	0.0002	-
*p* _13_	−0.00891	−0.00879	1.35	−0.00897	−0.008911	0.67	−0.00900	−0.00919	2.11
*q* _13_	−0.04407	−0.04418	0.25	−0.04197	−0.042464	1.19	−0.04328	−0.04353	0.58
*r* _13_	0.00079	0.00080	1.27	0.00065	0.000659	1.38	0.00058	0.00059	1.72
*θ* _y13_	1.83254	1.84138	0.55	1.45724	1.447243	0.69	0	0.0003	-
*θ* _xd13_	4.56376	4.60134	0.82	3.62178	3.641776	0.55	0	0.0002	-

Remark: The unit of deformation is m and that of the rotation angle is °.

## Data Availability

The authors confirm that the data supporting the findings of this study are available within the article. In addition, the data that support the findings of this study are available from the corresponding author, Jianwei Wu, upon reasonable request.

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
