# Peer review of "A New Nonlinear Spatial Compliance Model Method for Flexure Leaf Springs with Large Width-to-Length Ratio under Large Deformation"

_micromachines, 2022, doi:10.3390/mi13071090_

Round 1

Reviewer 1 Report

This work presents a large nonlinear spatial compliance model for flexure leaf springs of large width-to-length ratio. Large spatial deflection has been one of most challenging problems in compliant mechanism research, this work provides a solution for this problem. The model was verified by nonlinear finite element results and experimental results. Overall, it is a technically sound work. I recommend its acceptance after a minor revision.

1. Please remove the first paragraph in Introduction.

2. Please mention the following work on the similar topic:

Modeling large spatial deflections of slender beams of rectangular cross sections in compliant mechanisms, ASME Journal of Mechanisms and Robotics, 2021, 13(1): 011021

3. Figure 18: please show the deflection contours (using different colors) of the deformed FLS.

4. Eq. (15): please rewrite it as a standard boundary condition expression.

5. Please correct Ri in Eq. (32). 

Author Response

Dear reviewer:

Thank you for your all comments on my manuscript. I thanks for your high-quality review on my manuscript. I have revised my manuscript according to your comments. My response to your comments are as following:

  1. The first paragraph in Introduction is removed.
  2. The following work is referenced in Introduction:"Modeling large spatial deflections of slender beams of rectangular cross sections in compliant mechanisms, ASME Journal of Mechanisms and Robotics, 2021, 13(1): 011021"
  3. The different colors are used to show the deflection contours.
  4. The Eq. (15) is rewriten as a standard boundary condition expression.
  5.  The Ri in Eq. (32) is corrected.

Reviewer 2 Report

The manuscript describes the theoretical modeling and experimental verifications for the flexure leaf springs under large deformation. The Timoshenko beam theory was considered to include the shear effect for the beam with the high aspect ratio. The beam compliances have been derived by using the spatial chain model and verified via the simulation and experiments. Even though there is no original theory in this manuscript, it is worth introducing the scientific approach for the flexure modeling to the reader.

The current manuscript has only considered the Timoshenko beam theory. To improve the soundness of the research, I recommend authors to consider the higher order beam theory of Levinson such as the following paper. “Nghia-Huu Nguyen, Moo-Yeon Lee, Ji-Soo Kim, Dong-Yeon Lee, "Compliance Matrix of a Single-Bent Leaf Flexure for a Modal Analysis", Shock and Vibration, vol. 2015, Article ID 672831, 10 pages, 2015. https://doi.org/10.1155/2015/672831”

Author Response

Dear reviewer:

Thank you for your all comments on my manuscript. I thanks for your high-quality review on my manuscript. I have revised my manuscript according to your comments. My response to your comments are as following:

  1. The following  paper is referenced in the second paragraph of the Introduction:“Nghia-Huu Nguyen, Moo-Yeon Lee, Ji-Soo Kim, Dong-Yeon Lee, "Compliance Matrix of a Single-Bent Leaf Flexure for a Modal Analysis", Shock and Vibration, vol. 2015, Article ID 672831, 10 pages, 2015. https://doi.org/10.1155/2015/672831”
  2. The language of the revised manuscript is further polished by commissioning a professional language editing agency.

Reviewer 3 Report

The paper presents a commendable research work, proposing, a 6-DOF compliance model for Flexure Leaf Springs with large width-to-length ratio under large deformation. The theoretical work was verified through finite element simulations and after that validated experimentally.

There are some minor issues which should be addressed:

Please remove lines 28-36.

What software solution was used for finite element simulation?

Author Response

Thank you for your all comments on my manuscript. I thanks for your high-quality review on my manuscript. I have revised my manuscript according to your comments. My response to your comments are as following:

  1. The lines 28-36 is removed.
  2. The software was used for finite element simulation is ANSYS Workbench 18.0.
